# Association of fluid balance with mortality in sepsis is modified by admission hemoglobin levels: A large database study

Sandra M. Y. Tan⬤[1‡]*, Yuan Zhang[2‡], Ying Chen[3], Kay Choong See[4‡]*, Mengling Feng⬤[5‡]

1 Division of Advanced Internal Medicine, Department of Medicine, National University Hospital, Singapore, Singapore, 2 Ping An Healthcare Technology, Beijing, China, 3 Computational and Systems Biology, Genome Institute of Singapore, Singapore, Singapore, 4 Division of Respiratory and Critical Care Medicine, Department of Medicine, National University Hospital, Singapore, Singapore, 5 Saw Swee Hock School of Public Health, National University of Singapore and National University Health System, Singapore, Singapore

‡ SMYT and YZ are Joint Primary Authors. KCS and MF are Joint Senior Authors.
* sandra_tan@nuhs.edu.sg (SMYT); kay_choong_see@nuhs.edu.sg (KCS)

**Data Availability Statement:** The datasets analysed during the current study are available in the Medical Information Mart for Intensive Care-III

## Abstract

### Purpose

Sepsis involves a dysregulated inflammatory response to infection that leads to organ dysfunction. Early fluid resuscitation has been advocated by the Surviving Sepsis Campaign guidelines. However, recent studies have shown that a positive fluid balance is associated with increased mortality in septic patients. We investigated if haemoglobin levels on admission to the intensive care unit (ICU) could modify the association of fluid balance with mortality in patients with sepsis. We hypothesized that with increasing fluid balance, patients with moderate anemia (hemoglobin 7-10g/dL) would have poorer outcomes compared to those without moderate anemia (hemoglobin >10g/dL).

### Materials and methods

This retrospective study utilized the Medical Information Mart for Intensive Care-III (MIMIC-III) database. Patients with sepsis, as identified by the International Classification of Diseases, 9th, Clinical Modification codes, were studied. Patients were stratified into those with and without moderate anemia at ICU admission. We investigated the influence of fluid balance measured within 24 hours of ICU admission on 28-day mortality for both patient groups using multivariable logistic regression models. Subgroup and sensitivity analyses were conducted.

### Results

8,132 patients (median age 68.6 years, interquartile range 55.1–79.8 years; 52.8% female) were included. Increasing fluid balance (in L) was associated with a significantly decreased risk of 28-day mortality in patients without moderate anemia (OR 0.91, 95%CI 0.84–0.97, p = 0.005, at 6-hour). Conversely, increasing fluid balance was associated with a significantly increased risk of 28-day mortality in patients with moderate anemia (OR 1.05, 95% CI 1.01–

(MIMIC-III) repository, https://physionet.org/content/mimiciii/1.4/.

**Funding:** This study is partially supported by the National Research Foundation Singapore under its AI Singapore Programme (Award Number: AISG-100E-2020-055). The National Research Foundation Singapore did not have any additional role in the study design, data collection and analysis, decision to publish, or preparation of the manuscript. And the funder, Ping An Healthcare Technology, Beijing, China, provided support in the form of salaries for author Yuan Zhang, but did not have any additional role in the study design, data collection and analysis, decision to publish, or preparation of the manuscript. The specific roles of the author are articulated in the 'author contributions' section.

**Competing interests:** We declare a potential competing interest with regard to author Yuan Zhang's commercial affiliation with Ping An Healthcare Technology, Beijing, China. There are no further relevant declarations to make with regard to this commercial affiliation. This does not alter our adherence to PLOS ONE policies on sharing data and materials.

**Abbreviations:** CI, Confidence Interval; FB, Fluid Balance; GCS, Glasgow Coma Scale; Hb, Hemoglobin; ICD-9-CM, International Classification of Diseases, Ninth Revision, Clinical Modification; ICU, Intensive care unit; IQR, Interquartile Range; KNN, K-Nearest Neighbors; MIMIC-III, Medical Information Mart for Intensive Care III; OASIS, Oxford Acute Severity of Illness Score; OR, Odds Ratio; SIRS, Systemic Inflammatory Response Syndrome; SOFA, Sequential Organ Failure Assessment; STROBE, Strengthening the Reporting of Observational Studies in Epidemiology.

1.1, p = 0.022, at 24-hour). Interaction analyses showed that mortality was highest when haemoglobin decreased in patients with moderate anemia who had the most positive fluid balance. Multiple subgroups and sensitivity analyses yielded consistent results.

## Conclusions

In septic patients admitted to ICU, admission hemoglobin levels modified the association between fluid balance and mortality and are an important consideration for future fluid therapy trials.

## Introduction

Sepsis involves a dysregulated inflammatory response to infection that leads to organ dysfunction. It is associated with high mortality and morbidity in patients [1, 2]. In early sepsis, there is hypotension contributed by hypovolemia, vasodilatation and interstitial extravasation. The Surviving Sepsis Campaign guidelines [3] advocate early fluid resuscitation with 30mL/kg crystalloid for septic shock or sepsis-induced hypoperfusion. While fluid therapy in sepsis may stabilize the circulation and potentially be lifesaving, fluid overload is associated with development of tissue edema and worsened outcomes [4, 5]. Furthermore, positive fluid balances have been associated with increased mortality in septic patients, particularly after 24 hours of intensive care unit (ICU) admission [6] or with volumes larger than 5L during the first ICU day [7].

Although fluid administration can help expand intravascular volume and improve perfusion pressure, it is unclear if the iatrogenic hemodilution effect of resuscitation fluids could lead to poorer oxygen carriage and delivery causing fluid administration to paradoxically worsen tissue hypoperfusion. This effect could be made worse in patients with lower hemoglobin levels. A retrospective cohort study of the ARISE trial [8] found each litre of intravenous fluid administration was significantly but weakly associated with a reduction in haemoglobin concentration by 1–1.3g/dL at 24 hours and 72 hours of admission. Additionally, for each 1g/dL decrease in haemoglobin concentration during 72 hours of admission, duration of invasive ventilation, ICU stay and hospital length of stay were longer.

Anemia in septic ICU patients is a common finding and has multiple contributors [9] such as gastrointestinal losses [10], hemolysis [11], phlebotomy losses [12], proinflammatory cytokine inhibition of erythropoietin production [13] as well as fluid-loading leading to hemodilution [14, 15]. Two prior randomized trials—the FEAST trial in children [16] and the trial by Andrews *et al* in adults [17] demonstrated *increased* mortality with an early fluid resuscitation strategy. Both the FEAST and Andrews trials were conducted in patients who had moderate anemia (mean hemoglobin 7.1 g/dL in FEAST and mean hemoglobin 7.8 g/dL in the Andrews trial). Nonetheless, the results from both trials cannot be directly applied to ICU patients with lesser degrees of anemia. A subsequent analysis of the FEAST trial could not demonstrate an effect modification by anemia, likely due to the already low hemoglobin levels, showing uniformly deleterious effects of fluid balance across hemoglobin levels [18]. Unlike the FEAST trial analysis, our large database has allowed us to investigate the effect of fluid balance both in patients with and without moderate anemia.

We sought to investigate if hemoglobin levels at ICU admission would modify the association of fluid balance with mortality in sepsis among ICU patients with and without moderate anemia. We hypothesized that with increasing fluid balance in the first 24 hours of ICU

admission, those with moderate anemia (hemoglobin 7–10 g/dL) would have higher risk of 28-day mortality compared to those without moderate anemia (hemoglobin > 10 g/dL).

## Materials and methods

### Study design

A retrospective study was conducted using the Medical Information Mart for Intensive Care III (MIMIC-III) database [19], a single center ICU database which contains data of high temporal resolution including vital signs, medications, laboratory measurements, observations and electronic documentation for all patients admitted to Beth Israel Deaconess Medical Center ICUs between 2001 and 2012. Use of the MIMIC-III database has been approved by the institutional review boards of Beth Israel Deaconess Medical Center (Boston, MA) and the Massachusetts Institute of Technology (Cambridge, MA). This study is reported in accordance with the Strengthening the Reporting of Observational Studies in Epidemiology (STROBE) statement [20].

We included patients with sepsis, as identified by International Classification of Diseases, Ninth Revision, Clinical Modification codes (ICD-9-CM), aged above 16 years old who stayed in ICU for at least 6 hours. Exclusion criteria were as follows: patients who had been readmitted to the hospital or ICU, patients without fluid balance records in the first 6 hours after ICU admission, patients with erroneous fluid balance records, patients without hemoglobin records, patients with an admission hemoglobin<7 g/dL as well as patients who demised within 24 hours of ICU admission. We excluded patients with admission hemoglobin<7 g/dL as these patients would likely be given blood transfusions [3, 21, 22]. Patients with moderate anemia were defined as those with admission hemoglobin 7-10g/dL. Patients without moderate anemia were defined as those with admission hemoglobin>10g/dL. We chose a 10g/dL cut-off to be in-line with the threshold for blood transfusion for early goal-directed therapy in sepsis [23, 24].

The primary outcome was 28-day mortality in MIMIC-III database, which was acquired from the social security death registry. Fluid balance was calculated every 6 hours within the first 24 hours after ICU admission. Fluid input took into account oral fluids, intravenous fluids, blood products and medications. Fluid output took into account urine output. Total fluid balance was winsorized at the 1 and 99 percentiles to limit the effect of extreme values and to reduce the effect of possibly spurious outliers.

Demographic information included age and gender. Medical comorbidities were determined using Elixhauser codes at discharge [25]. We also used predictors of illness severity including Sequential Organ Failure Assessment score, Oxford Acute Severity of Illness Score [26] and minimum Glasgow Coma Scale score at admission. Values of body weight, hemoglobin, creatinine, white blood cell count, temperature, respiratory rate, heart rate and mean blood pressure at admission were also collected. Mechanical ventilation status was assessed at time of ICU admission and use of vasopressors was assessed during ICU stay.

### Statistical methods

Data were expressed as medians with interquartile ranges (IQR) for continuous variables, as these variables do not fit the normal distribution. Patient counts with percentages were reported for categorical variables. Continuous variables included age, weight, length of stay, Sequential Organ Failure Assessment score, vital signs, laboratory values and fluid balance. Categorical variables included gender, comorbidities, other clinical scores, use of mechanical ventilation on ICU admission and use of vasopressors.

To assess the similarity in baseline characteristics between those with and without moderate anemia, we used the Kruskal-Wallis test and Fisher's exact test for continuous variables and categorical variables respectively. Significance was considered at $p < 0.05$ level.

To assess the association of fluid balance with 28-day mortality, multivariable logistic regression was performed for each hemoglobin group. The effects were examined 6-hourly at 6 hours, 12 hours, 18 hours and 24 hours since ICU admission, and patients who were discharged from ICU before each time point were excluded. Models were further adjusted for confounders including age, gender, weight, admission lactate, creatinine, white blood cell count, temperature, respiratory rate, mean blood pressure, minimum Glasgow Coma Scale score, Sequential Organ Failure Assessment score, mechanical ventilation status at admission and comorbidities. Missing data of vital signs and lab tests were imputed using their median values. Sensitivity analyses were further conducted for three other imputation methods including K-nearest neighbors (KNN), imputed with mean values and with mode. We also performed propensity score matching to further adjust for confounding factors.

We plotted separate interaction graphs for patients with and without moderate anemia, to examine the association of mortality risk with both fluid balance and haemoglobin changes. We stratified fluid balance and the change of hemoglobin by tertiles, and calculated 28-day mortality for each tertile. We chose to present fluid balance data in tertiles in order to allow for a balanced number of patients in each group. In addition, Fisher's exact test was used to compare high mortality risk categories with other categories.

We performed subgroup analyses to investigate whether the effect of fluid balance on mortality risk would be modified among patients with congestive heart failure, patients with chronic kidney disease and patients who underwent mechanical ventilation at time of ICU admission.

To investigate whether receiving blood transfusion would modify the effect, we excluded patients who had received blood transfusions during their stay and performed multivariable logistic regression to investigate if the findings were consistent with our primary findings. In order to investigate the effects of fluid balance later in ICU stay, we further conducted sensitivity analysis till 48 hours of ICU admission, while excluding patients who demised at less than 48 hours of ICU admission. To investigate the effects of different hemoglobin levels [27], we conducted sensitivity analyses with modified definitions of moderate anemia as 7-8g/dL and 7-9g/dL. All statistical analyses were performed using R (version 3.4.3).

## Results

Of 46,520 patients in MIMIC-III, 12,636 were identified with sepsis. We excluded 188 patients who were younger than 16 years old, 1,836 patients with readmission records, 29 who stayed in ICU for less than 6 hours, 1860 patients without fluid balance records in the first 6 hours after ICU admission or with erroneous fluid balance records, 147 patients without hemoglobin measurements in the 24 hours after ICU admission, 212 with ICU admission hemoglobin measurement <7g/dL and 232 who demised within 24 hours of ICU admission, leaving 8,132 unique first admissions (Fig 1).

Baseline characteristics are summarized in Table 1. Based on admission hemoglobin, there were 2,469 (30.36%) patients with moderate anemia with median age 68.51 years (IQR 55.53, 79.46). 53.38% of moderate anemia patients were male compared with 44.46% of patients without moderate anemia. Additionally, moderate anemia patients had lower body weight (75.50 vs. 78.10 kg), and more comorbidities including hypertension (15.27% vs. 10.67%), congestive heart failure (27.22% vs. 22.78%), liver disease (11.79% vs. 7.36%), cancer (10.9% vs. 5.44%)

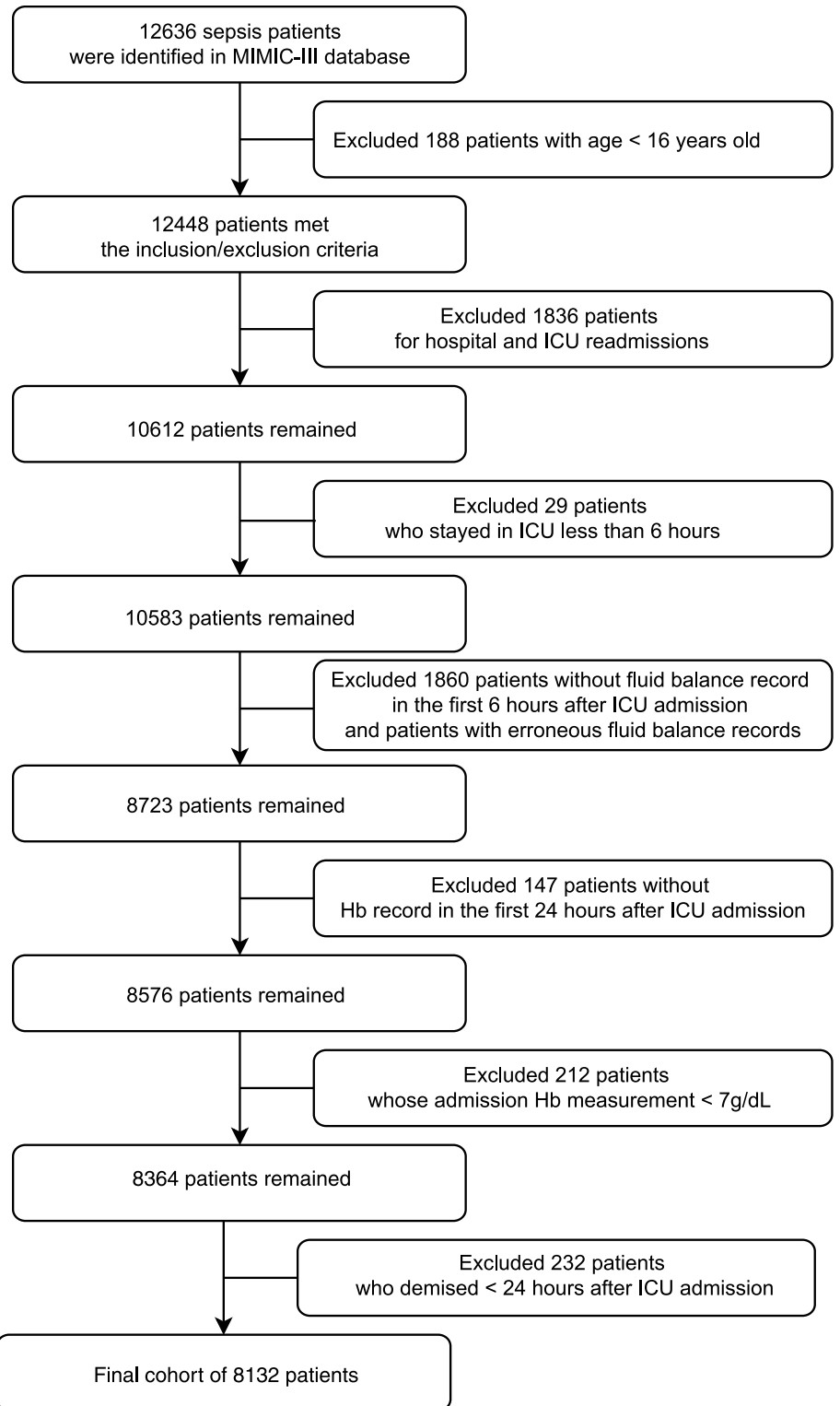

**Fig 1. Flow diagram of patient selection.** Flow diagram of cohort selection. Abbreviations: Hb = Hemoglobin; ICU = Intensive care unit; MIMIC-III = Medical Information Mart for Intensive Care III.

**Table 1. Baseline characteristics and outcomes.** Baseline observations and outcomes of included ICU patients with sepsis.

| | Moderate anemia | Without moderate anemia | p value |
|---|---|---|---|
| | n = 2469 | n = 5663 | |
| Age (median [IQR]), year | 68.51 [55.53, 79.46] | 68.64 [54.90, 79.91] | 0.837 |
| Gender = Male (%) | 1318 (53.38) | 2518 (44.46) | <0.001 |
| Weight (median [IQR]), Kg | 75.50 [63.40, 90.81] | 78.10 [65.00, 93.30] | <0.001 |
| Admission type (%) | | | 0.068 |
| Elective | 140 (5.67) | 384 (6.78) | |
| Emergency | 2329 (94.33) | 5279 (93.22) | |
| Length of stay (median [IQR]), Days | | | |
| Length of hospital stay | 13.44 [7.86, 23.45] | 12.34 [7.11, 20.69] | <0.001 |
| Length of ICU stay | 4.43 [2.12, 9.13] | 4.94 [2.39, 11.14] | <0.001 |
| Comorbidities (n, %) | | | |
| Hypertension | 377 (15.27) | 604 (10.67) | <0.001 |
| Congestive heart failure | 672 (27.22) | 1290 (22.78) | <0.001 |
| Liver disease | 291 (11.79) | 417 (7.36) | <0.001 |
| Cancer | 269 (10.9) | 308 (5.44) | <0.001 |
| Acquired immune deficiency syndrome | 16 (0.65) | 26 (0.46) | 0.355 |
| Chronic pulmonary disease | 505 (20.45) | 1175 (20.75) | 0.785 |
| Obesity | 140 (5.67) | 311 (5.49) | 0.787 |
| Chronic kidney disease | 536 (21.71) | 801 (14.14) | <0.001 |
| Clinical scores (median [IQR]) | | | |
| SOFA Score (median [IQR]) | 7.00 [4.00, 9.00] | 7.00 [4.00, 9.00] | 0.229 |
| SIRS Criteria (%) | | | 0.058 |
| < = 1 | 482 (19.52) | 1221 (21.56) | |
| = 2 | 699 (28.31) | 1578 (27.87) | |
| = 3 | 876 (35.48) | 2025 (35.76) | |
| = 4 | 412 (16.69) | 839 (14.82) | |
| Minimum GCS (median [IQR]) | 10.00 [6.00, 15.00] | 8.00 [3.00, 14.00] | <0.001 |
| OASIS Score (median [IQR]) | 38.00 [30.00, 47.00] | 41.00 [32.00, 47.00] | <0.001 |
| Lab tests (median [IQR]) | | | |
| Hemoglobin, g/dL | 9.10 [8.40, 9.60] | 12.10 [11.10, 13.40] | <0.001 |
| Lactate, mmol/L | 1.80 [1.20, 2.90] | 2.10 [1.40, 3.40] | <0.001 |
| Creatinine, μmol/L | 114.92 [70.72, 194.48] | 97.24 [70.72, 150.28] | <0.001 |
| eGFR, ml/min/1.73m2 (median [IQR]) | 52.03 [27.08, 88.66] | 57.12 [35.25, 87.49] | <0.001 |
| White blood cell count, x 10$^9$/L | 10.70 [7.10, 15.70] | 12.00 [8.40, 16.70] | <0.001 |
| Vital signs (median [IQR]) | | | |
| Temperature, °C | 36.72 [36.10, 37.39] | 36.70 [36.10, 37.39] | 0.961 |
| Respiratory rate, breaths per minute | 20.00 [16.00, 24.00] | 19.00 [15.00, 24.00] | 0.001 |
| Heart rate, beats per minute | 92.00 [79.00, 108.00] | 92.00 [78.00, 108.00] | 0.307 |
| Mean blood pressure, mmHg | 76.00 [65.67, 88.00] | 81.00 [70.00, 94.00] | <0.001 |
| Mechanical ventilation at admission (n, %) | 314 (12.71) | 836 (14.76) | 0.016 |
| Use of vasopressors (n, %) | 1143 (46.29) | 2723 (48.08) | 0.144 |
| Total fluid balance in first 24 hours of ICU stay (median [IQR]), L | 0.36 [-0.84, 2.07] | 0.18 [-1.10, 1.94] | <0.001 |
| Mortality (n, %) | | | |
| ICU mortality | 354 (14.30) | 679 (12.00) | 0.004 |
| 28-day mortality | 476 (19.30) | 913 (16.10) | 0.001 |

Abbreviations: GCS = Glasgow Coma Scale; ICU = Intensive Care Unit; IQR = Interquartile Range; OASIS = Oxford Acute Severity of Illness Score; SIRS = Systemic Inflammatory Response Syndrome; SOFA = Sequential Organ Failure Assessment.

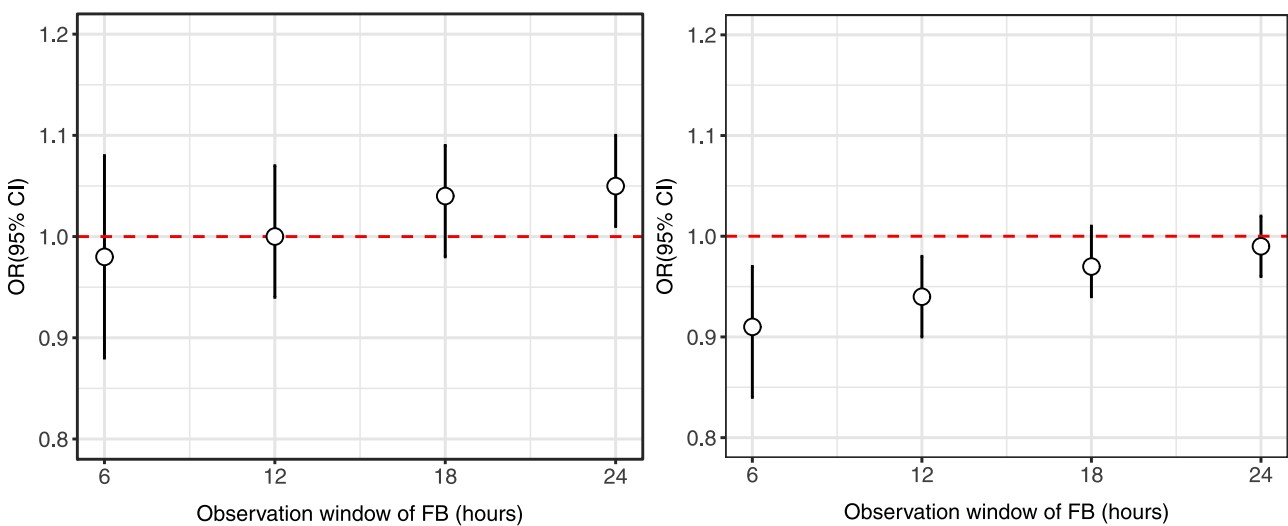

**Fig 2. Risk of 28-day mortality for sepsis patients at different observation windows after ICU admission.** (a) and (b) are for patients with and without moderate anemia respectively. Odds Ratio of each observation window represents the adjusted Odds Ratio of 28-day mortality derived from multivariable logistic regression. Abbreviations: FB = fluid balance; ICU = Intensive Care Unit; OR = Odds Ratio.

acquired immune deficiency syndrome (0.65% vs. 0.46%), obesity (5.67% vs. 5.49%) and chronic kidney disease (21.71% vs. 14.14%).

The median hospital length of stay of moderate anemia patients was higher (13.44 vs. 12.34 days), while ICU length of stay was shorter (4.43 vs. 4.94 days). Patients with moderate anemia also had a higher unadjusted 28-day mortality (19.3% vs. 16.1%). Median admission hemoglobin was 9.1g/dL (IQR 8.4–9.6) for moderate anemia patients and 12.1g/dL (IQR 11.0–13.4) for patients without moderate anemia. Patients with moderate anemia had lower admission lactate (1.8 vs. 2.1 mmol/L), lower white blood cell count (10.7 vs. 12 x $10^9$/L) and higher creatinine (114.9 vs. 97.2 μmol/L). 7577 (93.18%) patients had their first hemoglobin measured within 6 hours after ICU admission. 6,896 (84.8%) patients had more than 2 hemoglobin measurements in the first 24 hours. Hemoglobin change percentage was assessed for patients with more than 2 hemoglobin records, which was calculated as $(Hb_{last} - Hb_{first})/Hb_{first}$. More detailed illustration of hemoglobin measurements is included in S1 Fig.

In multivariable logistic regression analyses for patients with moderate anemia, increasing fluid balance (in L) was associated with increased risk of 28-day mortality at 24 hours after ICU admission (OR 1.05, 95% CI 1.01–1.1, p = 0.022) with median fluid balance of 1.78 L (IQR 0.78L–3.41L) (Fig 2a, Table 2). Median hemoglobin level overall showed a slight decrease from 8.8g/dL (IQR 7.9g/dL—9.6g/dL) at 6 hours after ICU admission, to 8.7g/dL (median IQR 8.1g/dL—9.4g/dL) at 24 hours after ICU admission. For patients without moderate anemia, increasing fluid balance was associated with decreased risk of 28-day mortality at 6 hours (OR 0.91, 95% CI 0.84–0.97, p = 0.005) and 12 hours (OR 0.94, 95% CI 0.9–0.98, p = 0.007) after ICU admission (Fig 2b, Table 2). Median fluid balance was 0.78L (IQR 0.38L–1.49L) at 6 hours and 1.19L (IQR 0.5L–2.33L) at 12 hours after ICU admission. Median hemoglobin level showed a decrease from 10.9g/dL (IQR 9.7g/dL– 12.2g/dL) at 6 hours after ICU admission, to 10.6g/dL (IQR 9.6g/dL– 11.8g/dL) at 24 hours after ICU admission.

28-day mortality for moderate anemia patients was calculated for each fluid balance tertile and according to hemoglobin change (S1 File) and represented on the interaction graphs in

**Table 2. Results of 28-day mortality for sepsis patients at different observation windows after ICU admission.** Logistic regression results of 28-day mortality for sepsis patients at different observation windows after ICU admission.

| Subgroup | Observation window | Patient Number | OR (95% CI) | Median fluid balance, L (Median [IQR]) | Median hemoglobin, g/dL (Median [IQR]) |
|---|---|---|---|---|---|
| **Moderate anemia patients** | 6 hours | 2469 | 0.98 (0.88, 1.08) p = 0.624 | 0.8 (0.35, 1.49) | 8.8 (7.9, 9.6) |
| | 12 hours | 2466 | 1 (0.94, 1.07) p = 0.97 | 1.15 (0.5, 2.33) | 8.7 (8, 9.5) |
| | 18 hours | 2459 | 1.04 (0.98, 1.09) p = 0.193 | 1.46 (0.66, 2.94) | 8.9 (8, 9.7) |
| | 24 hours | 2375 | 1.05 (1.01, 1.1) p = 0.022 | 1.78 (0.78, 3.41) | 8.7 (8.1, 9.4) |
| **Patients without moderate anemia** | 6 hours | 5663 | 0.91 (0.84, 0.97) p = 0.005 | 0.78 (0.38, 1.49) | 10.9 (9.7, 12.2) |
| | 12 hours | 5660 | 0.94 (0.9, 0.98) p = 0.007 | 1.19 (0.5, 2.33) | 10.9 (9.9, 12) |
| | 18 hours | 5644 | 0.97 (0.94, 1.01) p = 0.108 | 1.53 (0.68, 3.03) | 10.6 (9.7, 11.8) |
| | 24 hours | 5508 | 0.99 (0.96, 1.02) p = 0.405 | 1.84 (0.8, 3.52) | 10.6 (9.6, 11.8) |

In patients without moderate anemia on admission, increasing fluid balance was associated with decreased risk of 28-day mortality from the first 6 hours after ICU admission. Conversely in patients with moderate anemia on admission, increasing fluid balance was associated with increased risk of 28-day mortality at 24 hours after ICU admission. Abbreviations: CI = Confidence Interval; ICU = Intensive Care Unit; IQR = Interquartile Range; OR = Odds Ratio;

Fig 3. As shown, the 28-day mortality was significantly higher in patients with higher fluid balance and decreasing hemoglobin (dark pink pillar) than in patients with higher fluid balance but increasing hemoglobin or patients with lower fluid balance regardless of hemoglobin change (grey pillars): the highest mortality (29.25%) was found in fluid balance tertile 3

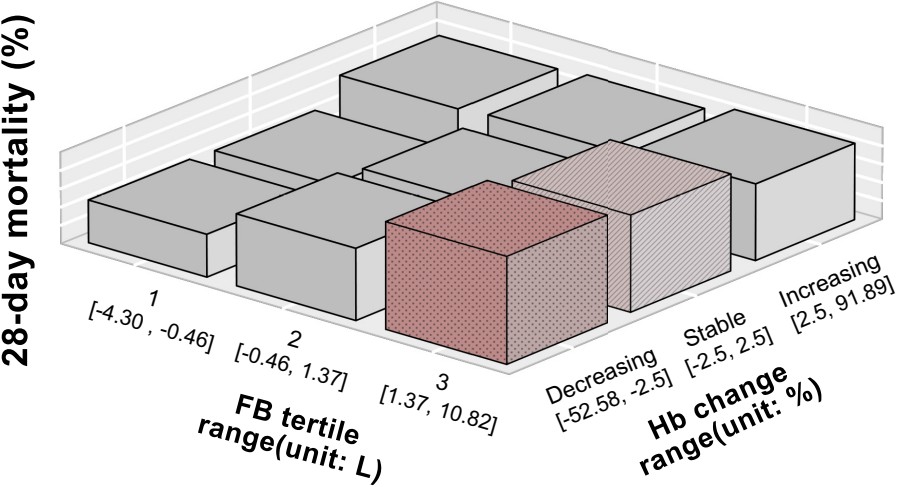

**Fig 3. 3D-bar plots showing 28-day mortality for different fluid balance and hemoglobin in patients with moderate anemia.** X-axis represents hemoglobin change, y-axis represents fluid balance and z-axis represents mortality at 28 days. For each graph, the pillar in dark red (dotted) represents the highest mortality. The grey pillars represent interactions with lower mortality (p<0.05). Pink pillar (striped) represents interactions in which no statistically significant difference from the highest mortality interaction was found. Mortality risk was highest when haemoglobin decreased in patients with moderate anemia who had the most positive fluid balance. Hemoglobin change% was calculated as 100 * (last hemoglobin record—first hemoglobin record) / first hemoglobin record. Fluid balance is stratified by the tertile of its value. Hemoglobin change was stratified to three groups: decreasing (< -2.5%), stable (±2.5%) and increasing (>2.5%). Abbreviations: FB = Fluid balance; Hb = hemoglobin.

**Table 3. Summary of findings in subgroup analyses and sensitivity analyses.**

|  | Findings | Related file |
|---|---|---|
| **Subgroup analysis** | • For patients with congestive heart failure, the results were in line with the main analyses for moderate anemia patients. In contrast, for patients without moderate anemia, there was no significant difference in risk of 28-day mortality.<br>• For patients with mechanical ventilation on ICU admission, no statistically significant difference in risk of 28-day mortality was found for patients with or without moderate anemia.<br>• For patients without mechanical ventilation on ICU admission, the results were in line with the main analyses for patients with and without moderate anemia.<br>• For patients with chronic kidney disease, no statistically significant difference in risk of 28-day mortality was found for patients with or without moderate anemia. | S2 Fig **and** S1 Table |
| **Sensitivity analyses for patients without blood transfusion** | In line with results of main analyses, magnitude of risk of 28-day mortality was higher in patients with moderate anemia compared to patients without moderate anemia. | S3 Fig **and** S2 Table |
| **Sensitivity analyses for extending observation window to 48 hours** | Results were consistent with main analyses | S4 Fig **and** S3 Table |
| **Sensitivity analyses for imputation method** | Results were consistent with main analyses | S5 Fig |
| **Sensitivity analyses for propensity score matching** | Results were consistent with main analyses | S2 File |
| **Sensitivity analyses for modified definitions of moderate anemia** | In line with results of main analyses, with moderate anemia defined as both 7-8g/dL and 7-9g/dL, no significant difference in risk of 28-day mortality was found in moderate anemia patients. Moreover, decreased risk of 28-day mortality was found in patients without moderate anemia. | S3 File |

Abbreviations: ICU = Intensive Care Unit

(1.37L–10.82L, median 3.01L IQR 2.07L–4.49L) and decreasing hemoglobin (-52.28%—< -2.5%, median -9.38%, IQR -14.29%–-5.81%) within 24 hours of ICU admission. The lowest mortality (11.69%) was found in fluid balance tertile 1 (-4.3L–0.46L, median -1.29L, IQR -2.01L–-0.85L) and stable hemoglobin (-2.5%–2.5%).

## Subgroup and sensitivity analyses

A summary of findings from our subgroup analyses and sensitivity analyses can be found in Table 3. Results remained consistent with findings of our main analyses.

## Discussion

The main and novel finding from our study was that among patients with moderate anemia, increasing fluid balance within the first 24 hours of ICU admission was associated with *increased* risk 28-day mortality. This is in contrast to the finding of *decreased* risk of 28-day mortality in patients *without* moderate anemia. Hemoglobin trend for all patients, with and without moderate anemia at admission, appeared to decrease through the first 24 hours of ICU stay, in line with studies done previously [28, 29].

For patients without moderate anemia (hemoglobin >10 g/dL) at ICU admission, increasing fluid balance was associated with decreased risk 28-day mortality in the first 24 hours of

ICU stay (Table 2, Fig 2). For these patients, in the early phase of critical illness, fluid expansion likely improved stroke volume by increasing both preload and contractility [30], and had a net benefit of increasing oxygen carriage to the tissues.

For patients with moderate anemia (hemoglobin 7–10 g/dL) at ICU admission, increasing fluid balance was associated with increased risk of 28-day mortality. This was demonstrated more clearly when we extended our analysis from the first 24 hours to the first 48 hours of ICU admission, and we found that increasing fluid balance was associated with increased 28-day mortality at 24 hours of ICU admission and beyond. The association of increasing fluid balance with higher mortality was more pronounced in patients with moderate anemia who also had decreasing haemoglobin (Fig 3, S1 File). In patients with sepsis, it is known that positive fluid balance leads to tissue edema. In the pulmonary system, this impairs oxygen diffusion and causes hypoxic pulmonary vasoconstriction [31] which leads to right heart strain. In the cardiovascular system, in the context of septic cardiomyopathy [32], increased filling pressures can further drive lung edema and acute lung injury. In the hepatic and renal systems, it causes liver and renal hypoxia respectively. In the liver, increased right ventricular and inferior vena cava pressure decrease the perfusion pressure gradient across the liver [4]. Similarly in the kidneys, fluid overload increases pressure within the renal capsule, decreasing glomerular filtration rate and thus renal perfusion [33]. These complications of fluid therapy extend to other critically ill patients including those with acute lung injury [34] and acute kidney injury [35]. While intravenous fluid administration could replenish intravascular volume depletion in the early hours of sepsis, those with moderate anemia would have already poor blood oxygen-carrying capacity and fluid administration could lead to iatrogenic hemodilution and paradoxically worsen tissue oxygenation, leading to poorer outcomes.

In our subgroup analyses of patients with congestive heart failure, increasing fluid balance was associated with increased 28-day mortality in the moderate anemia group (OR 1.09 at 24 hours, 95% CI 1.01–1.18, p = 0.03). Unlike our primary analysis, in patients with congestive heart failure, there was no association of increasing fluid balance with decreased 28-day mortality in patients without moderate anemia. A possible mechanism is the reduced preload volume required to optimize cardiac output on the Frank Starling curve in patients with congestive heart failure, compounded by the presence of septic cardiomyopathy causing left ventricular dysfunction [36]. The fluid volume and time to tipping the patient from an optimized oxygen delivery state to pulmonary edema are thus lower. In our subgroup analyses of patients with chronic kidney disease at ICU admission, similar to patients with congestive heart failure, there was no association of increasing fluid balance with a decreased 28-day mortality in patients without moderate anemia. In these patients, diminished cardiorespiratory reserves limits any benefit from fluid loading.

On the contrary, in our subgroup analysis of moderate anemia patients with mechanical ventilation at ICU admission, there was no significant difference in risk of 28-day mortality with an increasing fluid balance. We postulate that this is because patients on mechanical ventilation have reduced oxygen utilization and thus are better able to tolerate the loss of oxygen delivery that accompanies increasing fluid balance.

Subgroup analyses in patients without mechanical ventilation at ICU admission showed consistent trends with our primary results. (S2 Fig and S1 Table). Our subgroup analyses suggest that cardiorespiratory reserve is required to benefit from fluid loading. Any form of cardiorespiratory limitation such as congestive heart failure or chronic kidney disease may lead to loss of benefit from fluid loading and a potential for harm.

In our sensitivity analyses of patients without blood transfusion during ICU stay (S3 Fig and S2 Table), using imputation methods (S5 Fig) and with propensity score matching (S2 File), trends for risk of 28-day mortality remained consistent with our main analysis for patients with

and without moderate anemia. Consistency in results was also apparent in our sensitivity analyses with modified definitions of moderate anemia of 7-8g/dL and 7-9g/dL (S3 File).

The strengths of our study include our large sample of patients as well as use of data from a high-quality ICU research database that contains granular data including physiologic data and time-stamped treatment. In addition, our study analysed multiple subgroups in order to determine if such patients would show similar fluid balance versus hemoglobin relationships. To our knowledge, ours is the first study that looked into the relationship between the presence of moderate anemia on admission, versus the temporal association of fluid balance on mortality in sepsis.

We acknowledge several limitations. Firstly, patients were not randomized thus confounding may exist. We attempted to correct for this with multivariable analyses adjusting for potential confounders as well as with sensitivity analyses. Secondly, from our observational study, we cannot infer any causal relationships. Nonetheless, while confounding by indication is a possibility, deleterious effects of excess fluid have been well supported by multiple observational studies and randomized controlled trials [7, 33, 37, 38]. Thirdly, we did not analyse the types of fluids used in resuscitation and their impact on results, as the focus of this paper is to infer the effect modification of admission haemoglobin levels on the association between increasing fluid balance and mortality. Fourthly, we did not have data regarding events that took place prior to ICU admission including amount of fluid resuscitation as well as duration of time between sepsis onset and time of entering the ICU. However, the fluid balance recorded in ICU over 24 hours were remarkably similar in patients with and without moderate anemia. Fifthly, as medical comorbidities were determined using Elixhauser codes at discharge, we acknowledge that these diagnoses could have included complications during the index admission rather than pre-existing comorbidities. Adjusting for these late-onset complications would bias towards the null, and would not affect the interpretation of significant associations found in our study.

Controversies remain due to the complexity of care settings, pathophysiology of sepsis in various populations and the need for multiple interventions in an intrinsically heterogeneous patient population with sepsis. Our research has highlighted that septic patients with moderate anemia on ICU admission may have poorer outcomes with an increasing fluid balance. Current guidelines for septic patients do not account for hemoglobin levels. There is thus a need to account for baseline hemoglobin levels and changes in hemoglobin, for further studies on conservative versus liberal fluids in sepsis. Additionally, studies are needed to determine critical hemoglobin thresholds during fluid resuscitation in septic and possibly other critically ill patients. Physiological changes, such as changes in the immune system and endothelial function, in response to different fluid balances need to be elucidated. Additionally, there is a need for validation of clinical decision-making models that can achieve integration of physiological parameters into a patient-specific context. To illustrate, for patients who are anemic at baseline, careful tracking of hemoglobin may be important during large volume fluid resuscitation. Clinicians can consider monitoring perfusion and oxygen balance in patients with moderate anemia. Continuous methods of hemoglobin trending [39–42] such as use of non-invasive SpHb will need to be studied, as repeated hemoglobin blood assays can only be done at intervals and can exacerbate the problem of anemia.

## Conclusions

In septic patients admitted to ICU, moderate anemia appeared to modify the relationship between fluid balance and mortality. Admission hemoglobin levels would thus be an important consideration for future prospective trials of fluid therapy in patients with sepsis.

## Supporting information

**S1 Fig. Distribution of hemoglobin measurements.** (a) Distribution of Hb measurements. (b) Time of Hb measurement after ICU admission (Hours). (c) Number of Hb measurements per patient in the first 24 hours after ICU admission. (d) Distribution of Hb change (%) in the first 24 hours after ICU admission. Hb change (%) was calculated as (*admission Hb- last Hb in 24 hours)/admission Hb*, where Hb measures after blood transfusion were not taken into consideration. Abbrievations: Hb = hemoglobin.
(DOCX)

**S2 Fig. Visualization of regression results for subgroups of sepsis patients at different observation windows after ICU admission.** Logistic regression results for subgroups. The ORs represent the risk of 28-day mortality for sepsis patients at different observation windows after ICU admission. Subgroups from a to h are as follows: moderate anemia patients with congestive heart failure, patients without moderate anemia with congestive heart failure, moderate anemia patients with mechanical ventilation at admission, patients without moderate anemia with mechanical ventilation at admission, moderate anemia patients without mechanical ventilation at admission, patients without moderate anemia without mechanical ventilation at admission, moderate anemia patients with chronic kidney disease, patients without moderate anemia with chronic kidney disease. For moderate anemia patients with congestive heart failure, an increased risk of 28-day mortality was observed at 24 hours (OR 1.09, 95% CI 1.09–3.42, p = 0.03) with increasing fluid balance (median 1.47L, IQR 0.76L–3.32L). For patients without moderate anemia and with congestive heart failure, there was no significant difference in risk of 28-day mortality (OR 1.01, CI 0.95–1.06, p = 0.737 at 24 hours) with increasing fluid balance. For moderate anemia patients with chronic kidney disease, there was no significant difference in risk of 28-day mortality (OR 1.09, 95% CI 0.98–1.21, p = 0.124 at 24 hours) with increasing fluid balance. For patients without moderate anemia and with chronic kidney disease, there was no significant difference in risk of 28-day morality (OR 1.06, 95% CI 0.97–1.16, p = 0.201 at 24 hours) with increasing fluid balance. For moderate anemia patients and without mechanical ventilation at ICU admission, an increased risk of 28-day mortality was observed at 24 hours (OR 1.08, 95% CI 1.03–1.13, p = 0.003) with increasing fluid balance (median 1.72L IQR 0.76L – 3.27L). For patients without moderate anemia and without mechanical ventilation on admission, there was a decreased risk of 28-day mortality (OR 0.9, 95% CI 0.83–0.97, p = 0.009 at 6 hours; OR 0.94, 95% CI 0.89–0.98, p = 0.011 at 12 hours) with increasing fluid balance (median 1.15L, IQR 0.49L – 2.25L at 12 hours). For moderate anemia patients with mechanical ventilation on ICU admission, there was no significant difference in risk of 28-day mortality (OR 0.96, 95% CI 0.85–1.07, p = 0.45 at 24 hours) with increasing fluid balance. For patients without moderate anemia and with mechanical ventilation on ICU admission, there was no significant difference in risk of 28-day mortality (OR 0.97, 95% CI 0.9–1.04, p = 0.429 at 42 hours) with increasing fluid balance. Abbreviations: CHF = Congestive heart failure; CKD = Chronic kidney disease; FB = Fluid balance; ICU = Intensive Care Unit; MV = Mechanical ventilation; OR = Odds Ratio.
(DOCX)

**S3 Fig. Visualization of sensitivity analyses results for patients without blood transfusion in their ICU stay.** Logistic regression results for sepsis patients without blood transfusion. The ORs represent the risk of 28-day mortality for sepsis patients who did not receive blood transfusion in their ICU stay, at different observation windows after ICU admission. (a) and (b) show the regression results for patients who did not undergo blood transfusion, with and

without moderate anemia, respectively. Magnitude of risk of 28-day mortality was higher in patients with moderate anemia (OR 1.17, 95% CI 1.04–1.31, p = 0.009 at 24 hours) compared to patients without moderate anemia (OR 1.05, 95% CI 1–1.1, p = 0.046 at 24 hours) with increasing fluid balance. For moderate anemia patients with congestive heart failure, there was increased risk of 28-day mortality at 24 hours (OR 1.28, 95% CI 1.02–1.62, p = 0.036). For patients without moderate anemia with congestive heart failure, there was no significant difference in risk of 28-day mortality with increasing fluid balance. For moderate anemia patients without mechanical ventilation on ICU admission, there was increased risk of 28-day mortality at 18 hours (OR 1.17, 95% CI 1.01–1.36, p = 0.038) and 24 hours (OR 1.21, 95% CI 1.06–1.37, p = 0.004) with an increasing fluid balance. For patients without moderate anemia without mechanical ventilation on ICU admission, there was no significant difference in risk of 28-day mortality with an increasing fluid balance. Abbreviations: FB = Fluid balance; ICU = Intensive Care Unit; OR = Odds Ratio.
(DOCX)

**S4 Fig. Visualization of sensitivity analyses results extending to 48 hours of ICU admission.** Logistic regression results for extending to 48 hours of ICU admission. The ORs represent the risk of 28-day mortality for sepsis patients at different observation windows after ICU admission. (a) and (b) show the regression results for patients with and without moderate anemia respectively. There was an increased risk of 28-day mortality at 24 hours of ICU admission and beyond (OR 1.05, 95% CI 1.01–1.1, p = 0.027 at 24 hours; OR 1.08, 95% CI 1.04–1.11, p<0.001 at 48 hours) with increasing fluid balance in patients with moderate anemia. In patients without moderate anemia, there was a decreased risk of 28-day mortality (OR 0.87, 95% CI 0.81–0.94, p<0.001 at 6 hours; OR 0.92, 95% CI 0.87–0.96, p<0.001 at 12 hours; OR 0.95, 95% CI 0.92–0.99, p = 0.013 at 18 hours) with increasing fluid balance. Abbreviations: FB = Fluid balance; ICU = Intensive Care Unit; OR = Odds Ratio.
(DOCX)

**S5 Fig. Sensitivity analyses results for imputation methods.** Comparison of Odds Ratio of logistic regressions applying four imputation methods. The ORs represent the risk of 28-day mortality for sepsis patients at different observation windows after ICU admission. (a) and (b) show the regression results for patients with and without moderate anemia using four imputation methods including K-nearest neighbors, mean, mode and median imputation. Abbreviations: OR = FB = Fluid balance; Odds Ratio; KNN = K-nearest neighbors.
(DOCX)

**S1 Table. Regression results for subgroups of sepsis patients at different observation windows after ICU admission.**
(DOCX)

**S2 Table. Sensitivity analyses results for patients without blood transfusion in their ICU stay.**
(DOCX)

**S3 Table. Sensitivity analyses results extending to 48 hours of ICU admission.**
(DOCX)

**S4 Table. Missing data values.** Missing data of vital signs and laboratory tests were imputed with their median values.
(DOCX)

**S1 File. 28-day mortality at different hemoglobin percentage change and fluid balance change.**
(DOCX)

**S2 File. Sensitivity analyses results after propensity score matching.**
(DOCX)

**S3 File. Sensitivity analyses for different definitions of moderate anemia.**
(DOCX)

**S4 File. Description of fluid balance.**
(DOCX)

## Author Contributions

**Conceptualization:** Sandra M. Y. Tan, Yuan Zhang, Kay Choong See, Mengling Feng.

**Formal analysis:** Sandra M. Y. Tan, Yuan Zhang, Ying Chen, Kay Choong See, Mengling Feng.

**Investigation:** Sandra M. Y. Tan, Yuan Zhang.

**Methodology:** Sandra M. Y. Tan, Yuan Zhang, Kay Choong See, Mengling Feng.

**Supervision:** Kay Choong See, Mengling Feng.

**Writing – original draft:** Sandra M. Y. Tan, Yuan Zhang, Kay Choong See, Mengling Feng.

**Writing – review & editing:** Sandra M. Y. Tan, Yuan Zhang, Ying Chen, Kay Choong See, Mengling Feng.

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
