## [Decision Letter · Decision Letter 0]

3 Mar 2021

PONE-D-21-00459

Association of fluid balance with mortality in sepsis is modified by admission hemoglobin levels: a large database study

PLOS ONE

Dear Dr. Tan,

Thank you for submitting your manuscript to PLOS ONE. After careful consideration, we feel that it has merit but does not fully meet PLOS ONE’s publication criteria as it currently stands. Therefore, we invite you to submit a revised version of the manuscript that addresses the points raised during the review process.

We look forward to receiving your revised manuscript.

Kind regards,

Aleksandar R. Zivkovic

Academic Editor

PLOS ONE

We note that one or more of the authors are employed by a commercial company: PingAn Healthcare Technology, Beijing, China

(2) Please also provide an updated Competing Interests Statement declaring this commercial affiliation along with any other relevant declarations relating to employment, consultancy, patents, products in development, or marketed products, etc.  

Reviewers' comments:

Reviewer's Responses to Questions

**Comments to the Author**

1. Is the manuscript technically sound, and do the data support the conclusions?

Reviewer #1: Partly

Reviewer #2: Yes

Reviewer #3: Partly

2. Has the statistical analysis been performed appropriately and rigorously? 

Reviewer #1: I Don't Know

Reviewer #2: Yes

Reviewer #3: No

3. Have the authors made all data underlying the findings in their manuscript fully available?

Reviewer #1: Yes

Reviewer #2: Yes

Reviewer #3: Yes

4. Is the manuscript presented in an intelligible fashion and written in standard English?

Reviewer #1: No

Reviewer #2: No

Reviewer #3: No

5. Review Comments to the Author

Reviewer #1: The manuscript seeks to elucidate the impact of admission value of hemoglobin on the association between fluid balance and 28-day mortality in sepsis.

Major concerns:

1. Hemoglobin concentrations are referred to as "admission values", but time points for blood sampling are not specified. Page 11, lines 197-198 exclusion criteria: "174 without hemoglobin measurements in the first 3 days", does that mean that admission hemoglobin value is analyzed sometime during the first 72 h? Please, clarify as it seems important in order to be able to interpret results. Add to/modify accordingly in the Methods, (Results), Discussion.

2. p 16, line 261. The choice to present data in tertiles could be discussed and explained. The spread within tertiles is wide.

3. p 8, line 140 - 141 "Total fluid balance was winsorized at the 0.5 and 99.5 percentiles to limit the effect of extreme values", Please, give rational for why are data winsorized?

4. Figure 3a. tertile 1 has a negative fluid balance with the range -31,25 - 0,61. The authors might wish to discuss and explain how such an extreme negative fluid balance is possible in early sepsis and how these data should be interpreted. In fig 3b range of hemoglobin change in the third tertile (with increasing hemoglobin) is 5 - 285,7 %. This does seem peculiar as the group was defined as non-anemic. Are data correctly reported?

5. p9, line 157-159: "Total fluid balance during ICU stay was summarized as a continuous variable in baseline statistics and treated as a dichotomous variable (positive fluid balance versus negative fluid balance) in further modelling." Please, give rational for dichotomizing the data. The spread/variance of fluid balance over the first 24 hours as depicted in Figure 3a seems extreme. How do the investigators explain negative fluid balance of 32 L?

6. The manuscript might benefit from general revision of the language.

Minor concerns:

p 6, line 76 "It is associated with high mortality and morbidity in survivors" please reformulate

p 6, line 82, please comment further on the two references mentioned.

p 6, line 94 "duration of ventilation, length of stay" please specify

p 7, line 110-111 "have poorer outcomes in the first 24 hours", please correct

Reviewer #2: Thank you for allowing me to review “Association of fluid balance with mortality….” By Sandra Tan et al. Here, the purpose was to examine whether low hemoglobin (Hb) levels are associated with a poorer outcome of sepsis treatment. There are undoubtedly many patients studied, 9,700, and the included data are from admission to the ICU only (not before), the fluid balance during the first 24 h, and the mortality 28 days later. The results hold that positive fluid balance was associated with decreased 28-day mortality if Hb levels were on the high side. The reverse was the case in patients with low Hb levels (< 100 g/L).

My problem is that I don´t know if we learn anything from the results. Hb levels can be modified by many factors in severe disease. Inflammation leads to capillary leakage that raises Hb. If septic shock has occurred spontaneous capillary refill can be expected to having reduced the Hb concentration. Patients in bad condition and a low arterial pressure are likely to receive more IV fluid than others, which further reduces Hb.

We don't know in what clinical situation the patients were in when the “admission Hb” was taken. Had the patients already received i.v. fluid a bolus 30 mL/kg? Had they low MAP, which causes hemodilution by capillary refill? Had they waited a long time before entering the ICU? The analysis gets more complicated as Hb varies greatly among populations, and even within populations. Normal range in this country is 110-165 g/L.

If the authors can clearly point out the clinical implication of their results I am willing to change my judgment. Could it be that infusing much fluid on the first day of care is beneficial is Hb is high? If so, it would at least be logical on a simple level in a complex disease

Reviewer #3: Tan et al have investigated the association between admission hemoglobin levels and positive fluid balance influenced mortality in ICU patients.

1. The article lacks certain novelty, and the results need to be revised deeply; firstly, the amount of fluid balance is not clear in the definition of variables. Early fluid resuscitation in different ICU groups must be adjusted according to patients’ hemodynamic parameters. The more positive fluid balance doses have a greater impact on the prognosis of patients. I will suggest that the author can make a stratified analysis between different fluid balance and different hemoglobin levels to clarify the influence of hemoglobin levels on the prognosis of ICU patients during fluid therapy.

2. The definition of moderate anemia with hemoglobin 7-10g/dl is not properly, for hemoglobin lower than 8g/dl may influence patients’ outcome. The author needs to stratify different levels of hemoglobin, not just roughly with 7-10g/dl.

3. ICU patients’outcome can influence by many different parameters. It is difficult to exclude the mixing factors using logistical regression, and the results lack credibility. In this paper, to confirm the connection between positive fluid balance and hemoglobin levels in mortality, all the confound factors need to be strictly matched. At the same time, the statistical analysis needs to be further improved.

6. PLOS authors have the option to publish the peer review history of their article (what does this mean?). If published, this will include your full peer review and any attached files.

Reviewer #1: No

Reviewer #2: **Yes: **Robert G. Hahn

Reviewer #3: **Yes: **Milin Peng

---

## [Author Response · Author response to Decision Letter 0]

15 May 2021

Please see attached file titled 'Response to Reviewers'.

---

## [Editor Report · Decision Letter 1]

19 May 2021

Association of fluid balance with mortality in sepsis is modified by admission hemoglobin levels: a large database study

PONE-D-21-00459R1

Dear Dr. Tan,

We’re pleased to inform you that your manuscript has been judged scientifically suitable for publication and will be formally accepted for publication once it meets all outstanding technical requirements.

Kind regards,

Aleksandar R. Zivkovic

Academic Editor

PLOS ONE

---

## [Editor Report · Acceptance letter]

27 May 2021

PONE-D-21-00459R1 

Association of fluid balance with mortality in sepsis is modified by admission hemoglobin levels: a large database study 

Dear Dr. Tan:

I'm pleased to inform you that your manuscript has been deemed suitable for publication in PLOS ONE. Congratulations! Your manuscript is now with our production department. 

Kind regards, 

on behalf of

Dr. Aleksandar R. Zivkovic 

Academic Editor

PLOS ONE